# Forecasting Outbreaks of Hantaviral Disease: Future Directions in Geospatial Modeling

**DOI:** 10.3390/v15071461

**Published:** 2023-06-28

**Authors:** Gregory E. Glass

**Affiliations:** G&B Analytics, LLC, Gainesville, FL 32608, USA; gglass@ufl.edu

**Keywords:** Orthohantaviruses, hosts, transmission dynamics, forecasting, species distribution models, host specificity, outbreak prediction

## Abstract

Hantaviral diseases have been recognized as ‘place diseases’ from their earliest identification and, epidemiologically, are tied to single host species with transmission occurring from infectious hosts to humans. As such, human populations are most at risk when they are in physical proximity to suitable habitats for reservoir populations, when numbers of infectious hosts are greatest. Because of the lags between improving habitat conditions and increasing infectious host abundance and spillover to humans, it should be possible to anticipate (forecast) where and when outbreaks will most likely occur. Most mammalian hosts are associated with specific habitat requirements, so identifying these habitats and the ecological drivers that impact population growth and the dispersal of viral hosts should be markers of the increased risk for disease outbreaks. These regions could be targeted for public health and medical education. This paper outlines the rationale for forecasting zoonotic outbreaks, and the information that needs to be clarified at various levels of biological organization to make the forecasting of orthohantaviruses successful. Major challenges reflect the transdisciplinary nature of forecasting zoonoses, with needs to better understand the implications of the data collected, how collections are designed, and how chosen methods impact the interpretation of results.

## 1. Introduction

Human diseases caused by Orthohantaviruses were recognized for decades prior to the actual identification [1,2] and characterization [3,4,5] of the agents. As such, they appeared suddenly, with little warning and allowed little time to prepare a public health response. As knowledge of the variety of Orthohantaviruses and their clinical presentations grew [6,7], the recognized geographic and host ranges of the agents expanded [8]. Advances in molecular biology allowed the identification of the diverse hantaviral hosts [9,10]. However, human disease primarily remains associated with mammalian hosts. Hence, the discussion of forecasting risk focuses on this subgroup of agents.

An early characterization of hantaviral diseases, regardless of the agent, was that each tended to be geographically restricted to one or a few host species and jumped to human populations sporadically within those regions. This occasionally resulted in (relatively) large numbers of human cases, while other times few or no cases were observed (e.g., [11]). Consequently, it was presumed that certain human activities had markedly increased the risks of spillover of Orthohantaviruses from the common host species in a region [12]. The leading environmental factors were assumed to generate permissive conditions for host populations to increase and for viral infection to spread among hosts, making spillover to humans more likely [13]. If true, and the environmental conditions were identified rapidly enough, it should be possible to intervene and mitigate the outbreaks, especially for agents spilling over from wildlife [14].

The characterization is consistent with previously proposed strategies to forecast vector borne or zoonotic disease outbreaks. This paper provides a general overview of outbreak forecasting and describes a specific application as an example. It then outlines several of the milestones which are inherent in the attempts to apply forecasting to Orthohantaviral disease outbreaks, milestones that need to be met for practical, population-level interventions to be implemented. Resolving these assumptions will improve the quality of forecasting.

Many of these assumptions reflect details of the basic ecology and biology of the viruses, involving their persistence and transmission under natural conditions. Others involve the epidemiology of detecting early aspects of spillover and outbreaks of zoonoses, while still others reflect practical aspects of environmental monitoring and study design used to build forecast systems. These assumptions are taken, in turn, and demonstrated by various efforts of researchers around the world working in specific systems. All assumptions build Topon the groundbreaking efforts of Dr Ho Wang Lee and his collaborators from their earliest work in trying to understand this problem [15]. 

## 2. Forecasting Outbreaks

Outbreaks of vector-borne or zoonotic diseases in human or domestic animal populations are often sporadic, though some, such as falciparum malaria tend to occur on a regular seasonal basis (even if the amplitude of the outbreak varies dramatically). The extent to which outbreaks are sporadic in space and time makes interventions challenging. In some cases, such as the spread of West Nile virus [16] and the 2015 outbreak of Zika virus [17] in the western hemisphere, they can be anticipated when they arise in one region and are spread by travel patterns of humans [18].

Currently, outbreaks are identified and then responded to using confirmed reports of diseased humans or domestic animals once a detectable threshold is reached. These delays produce burdens on local or regional health care and economic systems [14]. Methods for implementing forecasts presume there is a staging of population level interventions to intervene more cheaply and efficiently [14]. As summarized in (Figure 1), the advantage of hypothetical forecasting systems is to reduce the duration and magnitude of outbreaks and mitigate impacts on health care infrastructure. This relies on forecasting to assess the likelihood of continued emergence of the virus and identify new/additional surveillance needs. Current systems can maintain active surveillance for a limited number of agents and primarily react to cases after detectable thresholds are reached. This strategy introduces a race between the growth of the ongoing epidemic and introducing interventions to terminate the outbreak (Figure 1Top). The challenge is that under current conditions active surveillance is not performed for pathogens before case levels are established. If the zoonotic agent adapts to human hosts, the risk of a pandemic becomes elevated. 

Because of the unpredictability of the times and places of outbreaks, surveillance rarely occurs during the early appearance of human cases and outbreaks remain undetected until a caseload threshold is elevated. This strategy generates a characteristic increase and subsequent decline (epidemic curve) in the outbreak. Thus, with current improved diagnostic methods and alert health care monitoring, we have good enough specificity to rapidly identify causative agents triggering the outbreaks after they develop, but the sensitivity of detecting when and where they are likely to emerge lags the successes of reactive identification.

Forecasting aims to improve the sensitivity for increasing alertness by feeding the information to decision support systems that provide alternative strategies for responses. This combines traditional passive surveillance within the public health infrastructure to identify new agents of concern and integrates information on environmental conditions that support emergence. If the environment is suitable, surveillance modes change to actively target regions and populations for human cases in environmentally suitable regions (Figure 1Middle). The information sources which monitor suitable environmental conditions are integrated into decision trees that trigger further health responses. Once the thresholds are reached, policy makers are engaged to establish the types and extents of responses available to evaluate the reported risk (Figure 1Bottom).

This forecasting system is conceptually like the framework for seasonal weather forecasting [19] or Famine Early Warning Systems [20]. The normal state of monitoring is established and the mid–long term forecasts are descriptive and broad. They are not intended to be precise nor to prescribe specific interventions. The goal is to give analysts the lead time to focus attention, if needed. However, if condition thresholds are reached, then a near-term forecast, using more detailed data and analyses, is instituted.

These systems have been applied for several vector borne viral pathogens. They cover broad geographic areas and large numbers of potentially impacted people and domestic animals. For example, part of the California mosquito control system gathers a series of data on meteorological, environmental, and social conditions that are monitored to identify environments that are suitable for vectors. These are analyzed to evaluate mid-term risk for the emergence of several arboviral disease outbreaks, based on historical understanding of mosquito population responses. Downstream, the analyses are coupled with various sources of ongoing monitored vector population abundances and virus infection levels to establish whether there is a need for increasingly intensive monitoring or interventions [21]. 

Orthohantaviruses have many characteristics which make them suitable for forecasting—surveillance, modeling, and intervention can be integrated to target and reduce the disease risk for the affected human populations. Applying these general strategies, I further outline issues to clarify. These are related to various aspects of the virus and host biology, and how to monitor environmental conditions such that forecasting the risk of Orthohantavirus outbreaks can be developed in various regions of the globe.

## 3. Orthohantavirus Host Specificity

Originally, hantaviruses were considered host specific, productively infecting and being persistently maintained in a single [22,23] or a few, closely related, mammalian species [2]. This was especially true when they seemed restricted to muroid rodents [8]. In recent years, however, there has been increasing evidence that this picture is incomplete. Initially, there was a single recognized Orthohantavirus, Thottopolyam virus, that was amplified by conserved S-segment primers [24], not associated with rodents. This virus was from an insectivore and analyses suggested it was an outgroup for the genus. Later work showed numerous distinct viruses were in Insectivora [25] and Chiroptera [26]. More recent work has identified even more hantaviruses in nonmammalian hosts [27,28]. 

The initial focus on rodent species was partly driven by surveillance bias, in that cases of the disease in humans triggered the investigations to find the environmental sources, rather than broad surveys of organisms to identify what species might be hosts to the viruses, even if apathogenic to people. This, in turn, was influenced by a lack of technology needed to easily identify new members of the group that did not rely on methods such as tissue culture or isolation (e.g., [29]).

Researching the phylogenetics of hosts and viruses is an important strategy to find currently unrecognized hosts and Orthohantaviruses [8,30]. However, the general concordance of host and virus trees suggests that successful species ‘jumping’ occurs uncommonly [26,27,28,30]. If it were common, then one would expect the trees of common hosts and the viruses to show very little congruity based on a shared evolutionary history and instead would correspond to geographic overlap of host ranges. Phylogenetics might form a key first step in recognizing the outbreaks of unknown Orthohantaviruses by checking sister taxa of hosts with pathogenic viruses. Given the array of new viruses, priority might be directed towards those new hosts that are common and in close proximity to humans as the species most likely to drive spillover of new agents.

Host specificity becomes important for forecasts in two situations. First, if there is host specificity and a specific virus is tied to a single host, then there is little likelihood for geographically syntopic species to productively amplify the target virus within their own populations and spillover the virus to humans in other environmental situations. In this situation, forecasting depends on the ability to identify environmental predictors of a single host population dynamics. However, if a single Orthohantavirus species uses multiple host species for transmission to humans then the space-time range of risk will extend beyond the niche space of any one host. 

Second, whether an Orthohantavirus productively infects two or more hosts and how it is maintained influences the likelihood of spillover. This phenomenon acts as a Boolean logical ‘AND’ or a Boolean ‘OR’ operator. In the case of an ‘AND’ agent, the virus resides primarily in one host, but is sequentially transmitted to a second host species that is in closer proximity to humans. This requires suitable environmental conditions for both species as well as some space/time overlap for sequential transmission (Figure 2). Such a disease system is reported for *Yersinia pestis* in Java, where several *Rattus* species sequentially transmit *Yersina* from one host species, in areas less inhabited by humans, to new host species at lower elevations that tend to live in and around local human populations [31]. Such a chain of transmission may have a lower likelihood of occurrence than a Boolean ‘OR’ system.

In Boolean ‘OR’ systems, more than one host species can maintain and transmit virus to humans so that if environmental conditions are suitable for either host, viral spillover to humans and outbreaks could occur under a wider range of environmental conditions. The extent to which two or more Orthohantavirus hosts commonly transmit a single virus to human populations needs to be better established. However, this would be most evident in specious regions of the world, such as the U.S. Southwest where many species for muroid rodents have overlapping habitat tolerances and tend to show long distance movements [13]. Similarly, in South America, spillover of the Jabora virus is reported in an array of species in the rodent fauna in Paraguay [32].

## 4. Fitness Effects

Whether Orthohantavirus infection in individual hosts impacts the survival or fecundity (fitness) of animals is relevant to forecasting because the number of infectious reservoir hosts are directly associated with human disease risk [33]. Forecasting that uses host population members’ states by accounting for individual fates or changes in intensity of infections during an outbreak [34] requires state-specific estimates of survival/fecundity. Modeling using agent-based models (ABMs), or ordinary differential equations (ODEs) rely on these metrics. These models must assume details of infectious host fates compared with the uninfected individuals as this impacts the predicted prevalence of host infections.

Generally, the presumption has been that Orthohantaviruses have little measurable impact on survival or fecundity [35]. Among natural populations, this assumption can be traced to the longitudinal (mark-release) study of Seoul virus in *Rattus norvegicus* [36]. In that study, there were no significant differences in the recovery of marked and released infected and uninfected rats, over time. Necropsies of females also observed no difference in embryo counts for infected and uninfected rats. However, Douglass and colleagues [37] reported that *Peromyscus maniculatus* infected with Sin Nombre virus (as measured by antibody) were less likely to be recaptured/survive than uninfected mice. 

Evaluating fitness impacts is a major challenge for natural populations of cryptic species such as rodents, insectivores, and bats. Laboratory studies do not address this question as laboratory conditions are presumably designed to reduce health risks that impact animals. However, in natural, open populations the loss to sampling follow-ups is presumed to represent the death of individuals, rather than movement from the sampled region or sampling avoidance. Thus, the actual recovery of dead animals is rarely done, and this is needed to distinguish between the alternatives. 

For example, bats are volant and may travel substantial distances, especially after disturbance, such as that caused by sampling [38,39]. Even terrestrial small mammals move much farther than usually recognized during their dispersal from their natal sites. *Peromyscus* spp. [13] were observed to move multiple kilometers during the breeding season from higher, more vegetated regions to drier more seasonal habitats at lower elevations. Few field studies of small mammals monitor such broad geographic movements and assume that loss to follow up sampling equates to death. Similarly, Armien and colleagues [40] demonstrated the broad geographic movements of *Oligorizomys fulvescens*, the host of Choclo virus, that was only identified by an extended sampling design (covering square kilometers) in Panama.

If transmission of virus among host population members is associated with a subclass in the population, and infection predominates in that subclass of hosts (e.g., subadult and adult male hosts) then infection and ‘apparent disappearance’ from the study area may be conflated. This generates a false association between infection and ‘decreased survival’. Dispersal from local regions is often associated with seasonal breeding in male wild rodents [41].

## 5. Orthohantavirus Modes of Transmission

The initial identification of Hantaan virus as a Bunyavirus led to studies to identify arthropod vectors [6,42,43,44]. However, even at a very early stage, it was recognized, by transmission to humans in laboratory settings, that arthropod vectors were not required, and that aerosol transmission was a likely source of contagion [45,46,47]. Later work showed viral nucleic acids were present in most organs of infectious hosts and excreted or secreted in most body fluids and feces [48]. These observations became critical in viewing Orthohantaviruses as occurring by horizontal transmission because the only obvious clinical disease in laboratory animals was when neonates were infected [49]. Neonatal (72 h) rodents suffered significant clinical disease and death but inoculation of 14 day old or older animals did not result in overt disease [44,49,50]. This was reinforced by laboratory observations that offspring born to infected dams were protected from subsequent virus challenge by maternal antibodies until it waned at roughly one month of age [51]. 

Vertical transmission could permit viral persistence with a minimal population size of a pregnant, infectious female and her progeny. However, horizontal transmission requires a larger local host population size to maintain the virus in the population (threshold population size). This has ramifications for viral persistence at the host population level. There must be sufficient susceptible individuals contacting (by whatever means) viable virus to maintain the transmission cycle. If local populations are too small, by stochastic processes alone, the virus will become extinct even if the host population persists [52]. This creates a patchwork of local host populations with and without the virus [53]. 

If there are fitness effects from hantavirus infections in hosts (see above), this would exacerbate the patchwork of viral persistence across the landscape. Longitudinal studies of *Peromyscus maniculatus* in the U.S. Southwest, show repeated examples of host populations persisting at low levels without the virus being found [54]. Similarly, studies of *Sigmodon hispidus* in south Florida, USA, showed many (54% of *n* = 89 sites) sampled local populations without Black Creek Canal virus [55]. Thus, landscape structure and the rates of movement of infectious hosts between habitat patches will likely impact the local risk of infection. Forecasting that focuses on host population dynamics without considering the loss and recolonization of virus from host populations will predict ‘false positive’ outbreak risks.

There is one report of viral antigens in Sertoli cells and spermatocytes in the testes of *R. norvegicus* infected with Seoul virus, under laboratory conditions [56]. It is not evident if sexual transmission plays a role in the natural maintenance of the viruses. If it did, one might anticipate that the reported strong male bias in host infections would be less obvious than is commonly reported [41]. Strikingly, *R. norvegicus* is one host that shows little sex difference in the prevalence of infection [23,29].

The apparent chronic nature of the infection in hosts further amplifies the need for large host population numbers. In situations where hosts clear infection and become susceptible again, there are three sources of susceptible population members; individuals born into the population, susceptible immigrants, and recovered individuals that become susceptible once more. With chronic infection, this last group is removed from the potential pool of susceptible individuals. Given the typically short life span of most rodent and insectivore species (Chiroptera are the exceptions) chronic infection does not need to be life-long, but merely long term. It, then, becomes important to better understand the effect of Orthohantavirus infection in hosts and whether chronically infected hosts continue to shed virus continually, effectively control shedding, or shed intermittently [48,50]. 

This suggests that environmental factors that favor large, contiguous populations of hosts are more likely to serve as sites of viral persistence than conditions that keep the host populations at low levels. As such, host intraspecific behavior and geographic landscape structure will likely influence spatial patterns of transmission. It also implies that the foci of infection are likely to be limited on the landscape. If identified, it would make it easier to focus surveillance and monitoring.

No single mode of horizontal transmission within host populations is obvious [56]. Studies by Nuzum [57] showed that both aerosol and inoculation routes effectively infected laboratory rats with Seoul virus. Aerosols or fomites can be generated from both animals and materials that they contaminate. However, inoculation was substantially more effective (ID (50) of 0.003 pfu vs. 0.5 for aerosol) than aerosol. A natural analogue of the injection is a bite, as a virus can be recovered from saliva and salivary glands [44,48,53]. The presumption that biting was a plausible route of transmission within host species is traced to a single longitudinal study of Seoul virus in wild *Rattus norvegicus* [58]. In this study, rats that were bitten between captures were seronegative at first capture, and were significantly more likely to seroconvert compared with those that were not bitten. Later studies examined collected animals for obvious wounds thought to be caused by intraspecific aggression [59,60]. Such observations were used to explain the apparent age and sex associated biases in the acquisition of the virus [61]. The limitation of these cross-sectional studies, as with the dispersal/survival fitness challenge, is whether there are confounding factors that may confuse the association.

Orthohantaviruses have, historically, been considered as an animal–human transmission. This makes forecasting feasible. Multiple, early epidemiologic studies ruled out human–human transmission [62]. This is critical for forecasting because it provides the needed lag time for viruses to be amplified in host populations and for infection prevalence to reach large enough levels that spillover is likely to occur [63]. However, the potential that Orthohantavirus transmission becomes a human adapted virus would be of special concern [63,64]. The best evidence for this is reported for the Andes virus, with sporadic occurrences of human–human transmission [65,66]. Very close person-to-person contact in medical settings, among household members, or at intimate social events seems to be associated with these events as even casual household contacts did not have an elevated risk [66]. 

Modes of horizontal transmission among host population members play important roles in forecasting population kinetics in hosts when demography becomes significant. Viral persistence in aerosols, body fluids, and excreta, or bedding [67] may be influenced differentially by environmental conditions (e.g., temperature and humidity) compared to direct contact. This may influence the rate of transmission between infectious and susceptible population members and when thresholds for action will be triggered.

## 6. Host/Human Interactions

The intensity of the space–time overlap of infectious hosts and susceptible humans is presumably related to the likelihood of spillover [63] (Figure 2). All other things being equal, Orthohantaviruses maintained by hosts that live in relatively undisturbed environments are less likely to be transmitted to humans than viruses carried by synanthropic hosts, simply because of the relative rarity of contact with humans. If transmission requires the host species, these will be lower risk within these systems. The conditions in Panama and the Southwestern United States emphasize the phenomenon. Numerous viruses associated with various host species occur in both regions [68,69]. However, Sin Nombre virus, primarily associated with the *P. maniculatus* and Cholco viruses, primarily associated with *Oligoryzomys costaricensis*, are overwhelmingly associated with human infection [70,71]. Both host species are eurytopic and, importantly, commonly associate with human housing in rural areas, at least on a seasonal basis [40,67,72,73]. However, hantaviruses occurring in hosts who are associated with less human-modified environments represent reservoirs of virus pools that may ‘emerge’ as human populations expand the exploitation of previously unused habitats [74].

## 7. Environmental Correlates of Risk

Forecasting (Figure 1) the risk of disease extends the current programs by incorporating the leading environmental conditions which drive human risk and then monitoring those conditions [75]. Environmental conditions can be monitored by a combination of in situ and remote sensing systems [76,77]. Environmental monitoring requires a spatial and temporal resolution that is consistent with the dynamics of virus transmission within host populations. If Orthohantaviruses are predominantly transmitted horizontally among hosts, and individuals remain infected (if not infectious) for life, then the larger host populations only require the environmental data resolutions to capture a spatial resolution slightly smaller than that needed to maintain local transmission and a temporal resolution somewhat shorter than the expected lifespan of the host. 

Current remotely sensed satellite data have a sufficient space/time resolution to meet these needs [78]. Converting the data gathered by local ground stations and imagery into measures of environmental conditions that drive viral spillover, however, is conceptually challenging because it relies on knowing the range of environmental conditions suitable for the host species. Some of this can be done from the diagnostics for estimating the distributions of host species (see below).

Sometimes the environmental data have already been collected and converted into products. The assumption is that the products, developed for one task, are appropriate for surveillance. This assumption needs to be validated. For example, the Normalized Difference Vegetation Index (NDVI), is derived as a normalized difference in the reflections of light energy for visible and near infra-red portions of the electromagnetic spectrum [78]. The NDVI accurately shows trends that monitor vegetation types and health [79]. However, when applied to the historical outbreak of Sin Nombre virus in the U.S. Southwest there was no clear demarcation in the NDVI that was associated with increasing human disease [80]. This was likely due to the broad range of habitats that *P. maniculatus* occupied in the region [78,80,81].

Similarly, land use—land cover (habitat) data products are widespread, but the categories used (ignoring misclassification issues) may not be relevant to host environmental use [81] because the host does not perceive the differences in classes that are used for the product. Alternatively, spectral signatures can be created from remote sensing data before it is converted to a product. These signatures associate satellite data at training sites with the tasked outcome [80,81]. Unlike the NDVI, which has gone through numerous evaluations in the remote sensing scientific literature, there appears to be little evidence that these other generated data products are appropriate for forecasting zoonotic agent changes, either for the applied classifications, or in a space/time resolution. Rather, they have been applied with little evaluation of their accuracy.

There are numerous new sensor systems being developed that may improve our understanding of how environments drive Orthohantavirus dynamics in host species [76,77,78,79]. The evaluation and calibration of these systems should be an area of active investigation to establish their increased utility for zoonotic pathogen outbreaks. 

## 8. Forecasting Orthohantavirus Risk

Risk forecasts extrapolate (into the future) for unmonitored geographic locales using a relatively small database that monitors the locations of viral occurrence/prevalence or sites with hosts (or human disease). These are associated with more extensive environmental databases that have some relationship to viral amplification [75]. The approaches to establish the relationships and the graphic presentations are Species Distribution Models (SDMs). The SDMs show the regions where viral activity is expected to be elevated [82,83]. 

Early attempts with Orthohantaviruses, their hosts [84,85], or human outbreaks [80] demonstrated the feasibility of identifying general regions of high and low risk [84], that substantially reduced the area and populations thought to be at risk (generating the efficiency of design). These models also could incorporate temporal changes in human disease risk [13,82]. 

The methods are intended to extend local studies to future, broader geographic areas. However, they also can be used for historical analyses to better understand the emergence of disease. For example, Glass and colleagues [80] used the originally recognized the outbreak of HPS in 1992–1993 in the U.S. Four Corners region to develop a spectral signature for locations of increased risk. However, laboratory studies showed that Sin Nombre virus existed prior to the human disease being recognized in the region [8]. This led to the question as to why the outbreak in 1992–1993 was the first report of human spillover and whether previous, unrecognized outbreaks may have occurred. Retrospective analyses of satellite imagery of the same region as the 1992–1993 outbreak showed environmental signatures similar, if not more geographically extensive and severe, to 1988, during the previous El Niño Southern Oscillation event (Figure 3). This would imply an unrecognized outbreak of the pulmonary disease occurred in 1988–1989 in the San Juan, McKinley and Cibola counties of western New Mexico and Apache county of eastern Arizona. 

The analytical methods associating health outcomes with environmental parameters have received extensive investigation during the past few decades [86,87]. Diverse strategies to establish the associations have been proposed. Some of these are heavily influenced by statistical methods for pattern detection [88] while others are more driven by biological relationships between hosts and their environments [89]. Regardless, all methods make assumptions about the data that may be incorrect and influence results to often unknown extents [87]. Because the various methods often differ in their assumptions [90] and no method always generates the ‘best’ model, the strategy is to use various models and combine the outcomes. The results produce an overall superior product compared to any one single approach [87]. Once the various models are developed, they are merged into an Ensemble model to identify areas of agreement, as well as the range of disagreement among the methods [91,92]. In principle, these areas of disagreement create ancillary hypotheses to further explore the host/virus biology.

Given the various population factors (abundance, sex, age composition) affecting viral amplification (see above), as well as the interpretation of the meaning of the data, it is a challenge to combine results that differ in their scale of action into a single SDM. If environmental correlates are sufficiently strong, they may capture many of the lower-level risk factors [85]. Alternatively, it may be necessary that the databases explicitly incorporate details of local host populations into the SDM. There was a single attempt to integrate host features at individual, population, and local environment levels (as well as nearby environmental conditions) into a spatially explicit prediction for the Black Creek Canal virus (BCCV) prevalence in *Sigmodon hispidus* throughout South Florida [83]. This study used empirical Bayesian estimation to generate local prevalence estimates of BCCV based on samples collected at an array of sites, with their associated biological characteristics at various levels of biological organization. Kriging was used to interpolate prevalence estimates in unsampled regions to the interior of the study. The method generated a map of the expected prevalence of BCCV for the region at 1 ha resolution. Importantly, it also mapped where the model’s estimates deviated from the estimated prevalence, such that further calibration could be performed.

## 9. Host/Virus Sampling Design

Sampling of the host/virus in the environment is the critical step for calibrating Ensemble forecasts by generating the monitored data sets that are fed into the analyses [90,92]. In some circumstances, such as during a case investigation or local outbreak, the goal is to implicate the agent and host in a restricted region. If there is sufficient sampling within these areas, finding environmental databases with high enough spatial resolution for the correct time period can be the challenge. However, regional, national, or international forecasting extrapolates information on host abundance/occurrence or virus prevalence/occurrence to broad regions of unsampled times/places and the extrapolation can be significantly impacted by how sampling is designed.

Much of the data used to find Orthohantavirus hosts relied on retrospective studies of materials collected for other purposes, such as museum collections [8]. As such, identifying (at least) viral sequences was sufficient to establish the occurrence of a specific virus in a specific location [8,9,10,28]. In many cases, these materials were collected originally to identify the ranges of mammalian species in certain regions. They were not designed to forecast viral occurrence in other locations where the species might be found. As such, they were ‘convenience’ samples for viral testing. Such samples introduce bias if used in forecasting studies [93]. 

Biased sampling is inherent in many natural history studies [8] and likely impacts forecasts of Orthohantaviruses (and other zoonoses). As such its significance depends on the question to be addressed. Despite recognizing the issue [86,87,94], and ongoing efforts to find solutions [95,96,97], few applied SDM studies incorporate the approaches [92]. However, observational epidemiology focuses on bias impacts in field studies. It has made extensive efforts to understand the sources, impacts, and solutions to biases in studies [98]. 

Overall, the challenge with biased studies is that they prevent, to an unknown degree, extending how we relate patterns in the collected sample to the source population from which samples were collected, and ultimately, to the population, as a whole [98]. This is exactly the goal of forecasting. The monitored populations produce information at a limited number of locations during a limited number of times. Forecasting attempts to accurately extend that to broader (unsampled) regions in the future. As such, prospectively gathered data for forecasts should identify and minimize sampling schemes that introduce bias [86,87]. 

The first step in designing forecasting survey studies is clarifying what questions the collections will address. Some rationale for the sampling frame should be explicitly outlined [98,99]. Forecasting requires adequate sampling throughout the range of environments where people may contact infectious hosts (excluding human-human transmission systems). This, ideally, includes at least a conceptual model of how environmental factors (broadly) are expected to drive the outbreak in the host populations. The complexity of ecosystem dynamics and human population interactions make this challenging. A qualitative approach used in observational studies uses Directed Acyclic Graphs (DAG) to identify major environmental factors that may confuse the interpretation of the observed results, as well as providing approaches to overcome them. This graphical approach recognizes two major categories of intervening environmental/social factors that need to be identified and resolved: confounders and colliders [100]. Confounders are (environmental) factors associated with exposure (such as host occurrence) as well as infection (or human disease). Confounders can be dealt with by various means [98]. Colliders occur when the exposure and the outcome independently affect a third variable. In contrast to confounders, studies that inappropriately control the colliders introduce bias. The direction and magnitude of the bias cannot be predicted. A recent review of the use of DAGs in vector borne disease systems is outlined by Jackson and colleagues for West Nile virus transmission cycles [100]. In the ecology of Orthohantavirus host populations, as a simple example, population structure, and sex ratio, are both modulated by seasonal reproduction, and all may directly or indirectly impact the observed infection prevalence. Failure to adjust (when possible) for confounders may generate misleading interpretations. 

Previously generated data from field studies can be examined for hypothesized confounders and colliders but bias is unlikely eliminated. The goal of identifying bias in these studies is not to eliminate all sources and types of bias. Instead, recognizing biases allows researchers to better identify what conclusions can correctly be drawn from the samples and what further studies need to be undertaken. 

Several sampling approaches have been proposed to reduce bias in SDMs. Many involve reducing the number of times or number of animals that any site contributes to the model analysis. This reduces the data to ‘presence only’ location and population intensity is not considered. This approach is useful when the goal is to better establish the occurrence and range of viral hosts [101]. An alternative approach has been to adjust sampling bias by making it conditional on the collectors’ choices of sites. This can be done by identifying sites where sampling found other species, but the target species was absent. By assuming that the surveys were performed prior to knowing the host would be present in some areas but not others, there is an internal control for collector biases and the comparison of sites with and without the species provides some critical clues to host distribution [101]. 

Surveys such as those reported by Armien and colleagues ([102]; Figure 1 and Figure 4), show sites where *O. costaricensis* was captured, or not, or Cholco virus was present or absent when the host was present, using their survey protocols. Second order spatial statistics [103] could be applied to these data to establish whether there is spatial clustering of sites with the (infected) hosts compared to sites with either only uninfected hosts, or no hosts, at all. If significant spatial clustering is found, then there is a basis to further explore the patterns and new hypotheses can be developed to differentiate the effects of sampling bias from ecological effects.

Adjustment for historical sampling efforts which are useful for identifying species ranges, are not applicable for forecasting where the future density of (infectious) hosts is needed. As an interim step, establishing likely places and times that support infectious hosts [81] may be sufficient to target surveillance. However, this strategy assumes that the cross-sectional surveys adequately capture the range of environmental dynamics that drive the outbreaks. 

## 10. Future Directions

Ultimately, validating forecast models requires predicting in new areas/times and testing predictions in new locations to establish that the models perform as needed. An analogous situation arises with SDM models. Validation generated during model development often assesses the quality of the final model by partitioning the original data set and builds the model with a portion. This is followed by testing to determine how well the ‘held out’ sample fits the resulting model. This is repeatedly performed and used to measure model consistency. Such results establish the internal validity of the final model and determine if the results are overly susceptible to some part of the data set. Thus, internal validity is an important aspect of model development.

However, this is distinct from external validity that asks whether the model can accurately predict results if sampling is performed in new locations. Because forecasting tries to extend from local monitoring to broader regions at other times, the quality of the external validity of the forecast is critical. 

Machine learning and artificial intelligence (AI) show great potential to improve the efficiency of forecasts, especially with host/vector ranges [104]. AI especially offers huge potential for ingesting, integrating and outputting the forecasting methods that will be needed in the future. The major challenge will be for users to explicitly elucidate why the AI models produce the predictions that they make, and protocols are developed to assess their accuracy in conditions that differ from those used to generate the original analyses [105]. This will be the future equivalent of analyzing the internal vs. external validity of the classifiers [92]. 

Forecasting is a transdisciplinary research program. As such, it uses experts in many different disciplines, each with an understanding of the data in their own areas but often failing to grasp the implications of how their results are interpreted and applied by other, less expert users. For example, whether a host is infected or infectious (and how that is determined) is critical for our understanding of host/virus population kinetics in natural circumstances. However, infected and infectious are often used interchangeably in the literature of Orthohantavirus field studies [53]. Similarly, serological assays have a long history of use in these studies. They are used to understand how viruses are maintained and transmitted in populations, often using IgG antibody titer thresholds. However, chronic infection of Orthohantavirus hosts may result in circulating antibodies in the absence of shed virus [48,56]. Under these circumstances, infectious and resistant classes may be conflated. Even molecular techniques, such as PCR, may produce different results when used to identify hosts when compared with the recovery of the infectious virus [28,30].

Challenges with host ecology, such as survival vs. movement have been outlined above. Issues with study designs and their impacts on SDMs that inform ensemble models and are critical to forecasting are discussed elsewhere [92]. Thus, all steps in the forecasting chain have features whose implications need to be better understood by the users further down the system if forecasting is to be accurate, timely, and useful for improving population health.

In summary, several important future aspects of research could help make forecasting Orthohantavirus risk a practical application for public health.

Serology such as IgG ELISA should be able to demonstrate a correspondence between the infection status and the threshold titer used to identify hosts that have been infected. Serology (if accurate) represents prevalent infection data rather than incident infection. Longitudinal studies are best able to identify incidence rates. Incident data is the measure of current virus circulation critical to outbreaks.Virus recovery from hosts is the gold standard to establish if a species is a host. Full length sequences of Orthohantaviruses are critical in this regard [28]. To the extent it is practical, virus should also recovered. Establishing a vertebrate species as a host requires evidence for population level rates of infection so that one can distinguish a host species from an incidental spillover from a nearby host [12]. Viral sequencing among different host species is important to establish whether multiple vertebrates serve as sources for a single virus and would need to be monitored as potential sources of human infection.Host ecology studies need to expand beyond local population surveys to incorporate metapopulation structure of natural populations. This requires data on local birth-death rates as well as immigration-emigration data [53,55]. Differentiating death from dispersal will be key in understanding fitness effects on subpopulations of hosts most likely to disperse virus.Study design of survey methods that introduce biases need to be evaluated. Nearly every study can provide some important clue about virus persistence and distribution, but some designs are not appropriate for the conclusions that need to be reached.Ensemble SDMs are among the most efficient ways to extend local knowledge about viral levels in host populations to nearby humans. Ways to address biased sampling impacts on SDMs have received great attention but need to be better integrated into practical applications of vector borne and zoonotic disease forecasting.Future developments of machine learning and AI represent the best way forward to make forecasting practical in public health of these agents. It will be critical for developers to intimately understand when machine learning approaches improve analyses and whether AI is robust to deviations from the conditions for which the models were developed, and therefore if they will be of value.

## Figures and Tables

**Figure 1 viruses-15-01461-f001:**
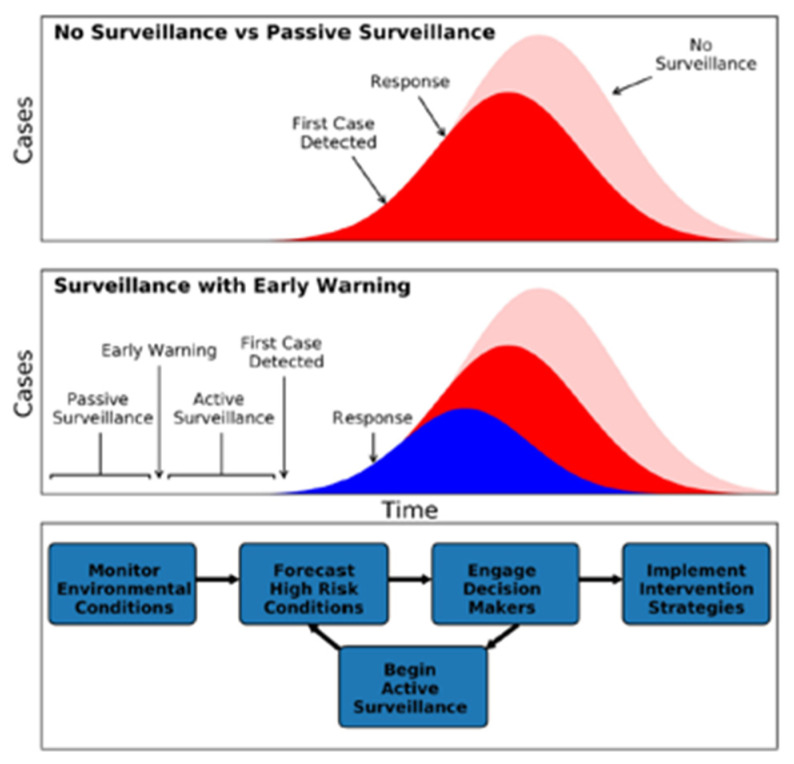
Graphic representation of outbreak detection and timing of response, (**Top**) without forecasting (=early warning), and (**Middle**) with forecasting and surveillance. Key decision steps with forecasting (**Bottom**). Used with permission (license number 5540270027357) [14].

**Figure 2 viruses-15-01461-f002:**
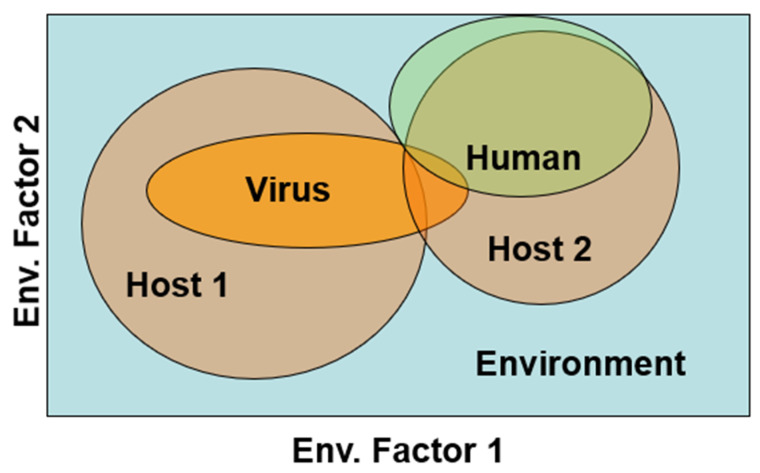
Niche overlaps of two hypothetical Orthohantavirus hosts and humans projected onto two environmental factors. The extent that the virus can be maintained by both hosts, independently, and transmitted to humans (Boolean OR) or must move sequentially from a host (#1) that rarely contacts humans, through a second host (#2) that broadly overlaps humans (Boolean AND) will impact the size of the human population at risk.

**Figure 3 viruses-15-01461-f003:**
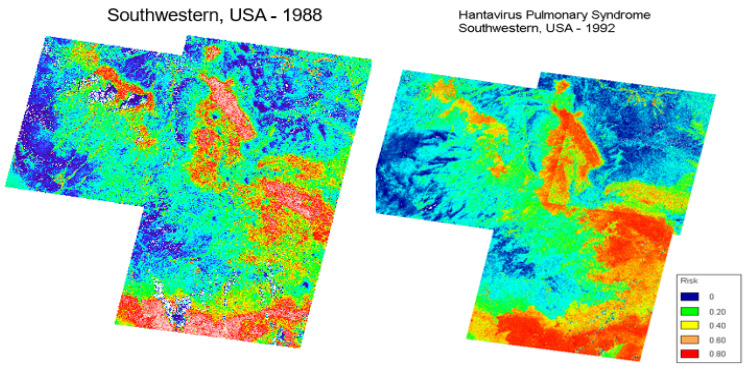
Satellite classification of Hantavirus Pulmonary Syndrome for the original outbreak region in the U.S. Southwest. Methods are described in [80]. Original landscape analysis is shown using Landsat TM imagery from the 1992 for 1993 outbreak (**right**). Suitability of the same region (**left**) four years prior to the recognized outbreak using the same color scale.

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
