# Peer review of "Forecasting Outbreaks of Hantaviral Disease: Future Directions in Geospatial Modeling"

_viruses, 2023, doi:10.3390/v15071461_

Round 1
Reviewer 1 Report
Line 70-78 and other: Once a disease is identified (especially a new one like HPS) ongoing surveillance depends on establishing a clinical case definition (modifiable as more information is gathered and specific laboratory tests are developed). Passive surveillance depends on collecting reported cases from various sources and is the most common method for gathering data. Active surveillance is resource intensive but a gold standard as it attempts to detect all cases including those that might not be identified by passive surveillance because of factors related to severity (e.g. requiring hospitalization) or with unusual presentation (HPS vs HPS with dominant cardiac syndrome).
Line 91: remove one ‘that’
Figure 1B assumes that surveillance already exists for an agent. What if unknown?
Line 115: Orthohantaviruses have many characteristics making them suitable for forecasting - 115 surveillance, modeling, and intervention WHICHexceptionally volant and may travel substantial distances, can be integrated to target and reduce the dis- 116 ease risk for affected human populations.
Line 137: such as tissue culture OR isolation (e.g., 29).
Line 201: Thus, the actual recovery of dead animals IS rarely done .
Line 203: exceptionally volant and may travel substantial distances, delete exceptionally.
Line 227-228: Neonatal (72 hours) 227 rodents suffered significant clinical disease and death but by day 14 of age, disease did 228 not occur (44, 49, 50). Sentence is unclear.
Line 256: The apparent chronic nature of infection in hosts, further amplifies the need for large 256 host population numbers. Not necessarily as long liuved chronically infected individuals can initiate infections among new cohorts of susceptibles. E.g. duration of HIV drives R0.
Line 332: Some of this can be done from the diagnostics from estimating distri- 332 butions of host species (see below). Remove second FROM replace with FOR.
Line 339-340: However, 339 when applied to the historical outbreak of Sin Nombre virus in the U.S. Southwest a 340 threshold for NDVI associated with human disease risk it was not evident (80). Needs more explanation about hypothesis tested.
Line 367: used the originally recognized outbreak add OF HPS.
Line 398: in a single SDM. INtO. (Note my keyboard is no longer producing capitol t).
Line 405: Bayesian estimation to estimate local prevalence estimates of BCCV…Lose one of the ‘estimates’.
Line 428: They were not designed to forecast viral occurrence other locations insert IN before other.
Line 430: Such samples introduce bias if used forecasting studies insert IN.
Line 643: 49. 49. Kim, G.R.; McKee, K.T., Jr. Pathogenesis of Hantaan Virus infection in suckling mice:… Delete 49 in reference.
Fine
Author Response
REVIEWER 1
RESPONSES ARE IN CAPS. First, my sincere thanks to the reviewer for taking the time to provide both thought provoking questions and issues with the writing that they should not have had to correct (my apologies).
Line 70-78 and other: Once a disease is identified (especially a new one like HPS) ongoing surveillance depends on establishing a clinical case definition (modifiable as more information is gathered and specific laboratory tests are developed). Passive surveillance depends on collecting reported cases from various sources and is the most common method for gathering data. Active surveillance is resource intensive but a gold standard as it attempts to detect all cases including those that might not be identified by passive surveillance because of factors related to severity (e.g. requiring hospitalization) or with unusual presentation (HPS vs HPS with dominant cardiac syndrome). I HAVE TRIED TO CLARIFY THIS VERY GOOD POINT BY THE REVIEWER IN LINES 70-90 AS I SEE NO FEASIBLE, TECHNICAL WAY TO FURTHER RAMP ACTIVE SURVEILLANCE (ALTHOUGH NEW TECHNOLOGIES MAY POINT A WAY). AS THE FIG 1, INDICATES, ACTIVE SURVEILLANCE EVEN WITHIN FORECASTING DOES NOT GET TRIGGERED UNTIL THERE IS EVIDENCE THAT THE ENVIRONMENT IS SUITABLE. THIS OBVIOUSLY RESTRICTS OUTBREAKS TO THINGS WE ALREADY KNOW. I TRY TO ADDRESS THE ‘UNKNOWNS’ IN THE FIG 1B BELOW.
Line 91: remove one ‘that’ DONE
Figure 1B assumes that surveillance already exists for an agent. What if unknown? I HAVE ADDED A SENTENCE (LINES 145-149). I WOULD SUGGEST THIS IS THE AREA WHERE PHYLOGENY OF POTENTIAL HOSTS (AND CONVENIENCE SAMPLES) ARE MOST SUCCESSFUL AT FINDING TRULY UNKNOWN AGENTS. THEN, I WOULD PRIORITIZE SEARCHS AMONG HOST SPECIES THAT ARE COMMON LOCALLY AND LIVE IN PHYSICAL/ECOLOGICAL PROXIMITY TO HUMAN POPULATIONS.
Line 115: Orthohantaviruses have many characteristics making them suitable for forecasting - 115 surveillance, modeling, and intervention WHICHexceptionally volant and may travel substantial distances, can be integrated to target and reduce the dis- 116 ease risk for affected human populations. DONE
Line 137: such as tissue culture OR isolation (e.g., 29). DONE
Line 201: Thus, the actual recovery of dead animals IS rarely done . DONE
Line 203: exceptionally volant and may travel substantial distances, delete exceptionally. DONE
Line 227-228: Neonatal (72 hours) 227 rodents suffered significant clinical disease and death but by day 14 of age, disease did 228 not occur (44, 49, 50). Sentence is unclear. HAVE TRIED TO CLARIFY THIS SENTENCE ON LINES 233-237. BRIEFLY, NEONATES GET SICK AND DIE. SUCKLING RATS (14 DAY OLDS) AND OLDER RATS THAT ARE INFECTED APPEAR DISEASE FREE AFTER INOCULATION. IF THIS REMAINS PROBLEMMATIC PLEASE LET ME KNOW.
Line 256: The apparent chronic nature of infection in hosts, further amplifies the need for large 256 host population numbers. Not necessarily as long liuved chronically infected individuals can initiate infections among new cohorts of susceptibles. E.g. duration of HIV drives R0. THIS IS AN EXCELLENT POINT. I WAS MORE THINKING ABOUT YANAGIHARA’S LAB WORK SHOWING THAT CHRONICALLY INFECTED RATS ONLY SPORADICALLY SHED HANTAVIRUS AND AS SUCH ARE MORE A ‘CHICKENPOX/ SHINGLES’ MODEL. IF ANIMALS ARE INFECTED (AND RESISTANT TO NEW INFECTION – SOMETHING I DON’T THINK ANYONE HAS CHECKED) THEN THEY REALLY ARE MORE IN THE ‘R SUBPOPULATION’ OF AN S-I-R MODEL AND AREN’T CONTRIBUTING TO INCIDENT INFECTIONS. THAT WOULD IMPLY WHAT INCIDENT INFECTIONS THAT DO OCCUR ARE MUCH MORE EFFICIENT THAN WE WOULD ESTIMATE IF WE INCLUDE THE ‘R’ CLASS AS MEMBERS OF THE ‘I’ CLASS. HOPEFULLY THE ADDITION OF LINES 270-272 CLARIFY MY THOUGHTS. IF THE REVIEWER STILL DISAGREES I WILL MODIFY THIS SECTION FURTHER.
Line 332: Some of this can be done from the diagnostics from estimating distri- 332 butions of host species (see below). Remove second FROM replace with FOR. DONE
Line 339-340: However, 339 when applied to the historical outbreak of Sin Nombre virus in the U.S. Southwest a 340 threshold for NDVI associated with human disease risk it was not evident (80). Needs more explanation about hypothesis tested. HAVE ADDED SENTENCE (LINES 350-352) TO (HOPEFULLY) CLARIFY THIS ISSUE. THE REASON, I BELIEVE, IS THAT DEER MICE EXTEND OVER SUCH A BROAD RANGE OF HABITATS THAT GREENESS, AT THE SPATIAL RESOLUTION OF LANDSAT, IS NOT ESPECIALLY MEANINGFUL AS A MEASURE OF PRODUCTIVITY OF THE HABITAT FOR DEER MICE.
Line 367: used the originally recognized outbreak add OF HPS. DONE
Line 398: in a single SDM. INtO. (Note my keyboard is no longer producing capitol t). DONE
Line 405: Bayesian estimation to estimate local prevalence estimates of BCCV…Lose one of the ‘estimates’. DONE
Line 428: They were not designed to forecast viral occurrence other locations insert IN before other. DONE
Line 430: Such samples introduce bias if used forecasting studies insert IN. DONE
Line 643: 49. 49. Kim, G.R.; McKee, K.T., Jr. Pathogenesis of Hantaan Virus infection in suckling mice:… Delete 49 in reference. DONE
Reviewer 2 Report
The manuscript describes different aspects that affect forecasting of hantaviral disease in time and space, including biological and ecological factors as well as technical/study design ones. All in all, it is a well-written and interesting synthesis of the forecast problem when it comes to disease ecology, with a focus on hantaviral diseases.
I understand it is a concept paper touching different aspects of a central problem in science, but there are parts where I'd wish the author would expand on, i.e., not only posing the problems but highlighting specific potential solutions and pathways to enhance our forecast exercises in order to make them more useful for timely decision making.
In the future directions, perhaps the manuscript would benefit from including new satellite data like SAR, hyper-spectral, very high resolution, etc. that could enhance environmental predictors. Also, cross-validation (described in the first paragraph of future directions section) followed by evaluation with external data (data not used for training) is already a rather common practice in SDM and machine learning. Perhaps this could also be integrated in the forecast section. Indeed, machine learning could also go there as there's an increasing number of papers using it in the field of disease ecology.
Last but not least, I'd suggest to include a conclusion and recap (perhaps as bullet points) of topics ordered by relevance, urgency or some criteria with the practical foreseen pathways to address them.
Minor:
- See attached pdf with fixed typos and minor grammar suggestions and comments.
- Quality of figures should be enhanced, they look blurry.
- Please see the unordered list of manuscripts that address the sampling bias problem in SDMs that I have included. There is a significant body of literature assessing different problems of forecasting with SDMs.

Author Response
REVIEWER 2
I have followed the reviewer’s comments in their entirety. I hope my edits are satisfactory and greatly appreciate the tremendous effort made by them to improve this work. For simplicity, I have added comments in CAPS here. Otherwise, the markups on the pdf were incorporated as suggested.
The manuscript describes different aspects that affect forecasting of hantaviral disease in time and space, including biological and ecological factors as well as technical/study design ones. All in all, it is a well-written and interesting synthesis of the forecast problem when it comes to disease ecology, with a focus on hantaviral diseases.
I understand it is a concept paper touching different aspects of a central problem in science, but there are parts where I'd wish the author would expand on, i.e., not only posing the problems but highlighting specific potential solutions and pathways to enhance our forecast exercises in order to make them more useful for timely decision making. I HAVE TRIED TO CAPTURE THE SOLUTIONS IN A BULLETED LIST IN SECTION 10 AS A CONCLUSION AND RECAP.
In the future directions, perhaps the manuscript would benefit from including new satellite data like SAR, hyper-spectral, very high resolution, etc. that could enhance environmental predictors. Also, cross-validation (described in the first paragraph of future directions section) followed by evaluation with external data (data not used for training) is already a rather common practice in SDM and machine learning. Perhaps this could also be integrated in the forecast section. Indeed, machine learning could also go there as there's an increasing number of papers using it in the field of disease ecology. HAVE ADDED SENTENCES (LINES 364-367) MENTIONING NEW SENSORS THAT CAN BE EXPLORED IF THET PROVIDE REALLY NEW INFORMATION (NOT JUST IMPROVEMENTS IN RESOLUTION). ON THE SECOND POINT, I AM HAPPY TO HEAR THAT NEW DATA ARE BEING APPLIED MORE COMMONLY FOR VALIDATION. I AM PRIMARILY RESPONDING TO RESEARCHERS FOCUSED ON APPLICATIONS AS THIS IS A DISAGREEMENT I HAVE HAD REPEATEDLY WITH FOLKS WORKING ON VECTOR BORNE PATHOGENS.
Last but not least, I'd suggest to include a conclusion and recap (perhaps as bullet points) of topics ordered by relevance, urgency or some criteria with the practical foreseen pathways to address them. THANK YOU FOR THIS SUGGESTION. HAVE PROVIDED RECAP IN SECTION 10 (LINES 558-589).
Minor:
- See attached pdf with fixed typos and minor grammar suggestions and comments. PROPOSED EDITS ARE MADE, AS SUGGESTED. I APPRECIATE (AND APOLOGIZE FOR THE ADDED WORKLOAD) THE EFFORT THE REVIEWER HAS MADE TO IMPROVE THE STRUCTURE
- Quality of figures should be enhanced, they look blurry. I HAVE SUGGESTED REDUCING THE SIZE OF THE FIGURES IN THE PAPER. I HAVE NO OTHER WAY TO SHARPEN THE IMAGES AND REDUCE PIXELATION.
- Please see the unordered list of manuscripts that address the sampling bias problem in SDMs that I have included. There is a significant body of literature assessing different problems of forecasting with SDMs. THIS WAS A WONDERFUL ADDITION THE REVIEWER HAS PROVIDED. I INCLUDED THREE OF THE MORE RECENT ONES IN THE MANUSCRIPT (#95-97). I VERY MUCH THANK THE REVIEWER FOR GENEROUSLY PROVIDING THESE.